# GDF-15 Deficiency Reduces Autophagic Activity in Human Macrophages In Vitro and Decreases p62-Accumulation in Atherosclerotic Lesions in Mice

**DOI:** 10.3390/cells10092346

**Published:** 2021-09-07

**Authors:** Aline Heduschke, Kathrin Ackermann, Beate Wilhelm, Lilli Mey, Gabriel Alejandro Bonaterra, Ralf Kinscherf, Anja Schwarz

**Affiliations:** Institute for Anatomy and Cell Biology, Department of Medical Cell Biology, Philipps-University of Marburg, 35032 Marburg, Germany; heduschk@students.uni-marburg.de (A.H.); Kathrin_Ackermann@gmx.net (K.A.); wilhelmb@staff.uni-marburg.de (B.W.); meyl@staff.uni-marburg.de (L.M.); gabriel.bonaterra@staff.uni-marburg.de (G.A.B.); ralf.kinscherf@staff.uni-marburg.de (R.K.)

**Keywords:** growth-differentiation factor-15, autophagy, atherosclerosis, p62

## Abstract

(1) Background: Growth differentiation factor-15 (GDF-15) is associated with cardiovascular diseases and autophagy in human macrophages (MΦ). Thus, we are interested in investigating autophagic mechanisms with special respect to the role of GDF-15. (2) Methods: Recombinant (r)GDF-15 and siRNA GDF-15 were used to investigate the effects of GDF-15 on autophagic and lysosomal activity, as well as autophagosome formation by transmission electron microscopy (TEM) in MΦ. To ascertain the effects of GDF-15^−/−^ on the progression of atherosclerotic lesions, we used GDF-15^−/−^/ApoE^−/−^ and ApoE^−/−^ mice under a cholesterol-enriched diet (CED). Body weight, body mass index (BMI), blood lipid levels and lumen stenosis in the brachiocephalic trunk (BT) were analyzed. Identification of different cell types and localization of autophagy-relevant proteins in atherosclerotic plaques were performed by immunofluorescence. (3) Results: siGDF-15 reduced and, conversely, rGDF-15 increased the autophagic activity in MΦ, whereas lysosomal activity was unaffected. Autophagic degradation after starvation and rGDF-15 treatment was observed by TEM. GDF-15^−/−^/ApoE^−/−^ mice, after CED, showed reduced lumen stenosis in the BT, while body weight, BMI and triglycerides were increased compared with ApoE^−/−^ mice. GDF-15^−/−^ decreased p62-accumulation in atherosclerotic lesions, especially in endothelial cells (ECs). (4) Conclusion: GDF-15 seems to be an important factor in the regulation of autophagy, especially in ECs of atherosclerotic lesions, indicating its crucial pathophysiological function during atherosclerosis development.

## 1. Introduction

Atherosclerosis, a chronic-inflammatory disease of the vascular wall, may lead to stroke, myocardial infarction or other cardiovascular events [1]. The progressive development of atherosclerotic plaques implies a deregulated apoptotic reaction of macrophages (MΦs), endothelial (ECs) and smooth muscle cells (SMCs), resulting in a formation of the necrotic lipid core, diminution of the fibrous cap and an inflammatory reaction [2].

Growth-differentiation factor (GDF)-15, a divergent and distant member of the TGF-β superfamily, identical to macrophage inhibitory cytokine-1 (MIC-1) [3], is widely distributed in adult tissues, being mostly expressed in epithelial cells, SMCs, adipocytes and MΦ [4]. Additionally, GDF-15 has been suggested as a biomarker for cardiovascular diseases [5], diabetes [6] and cancer [7]. Apparently, GDF-15 is a stress-inducible cytokine and is (up-) regulated by several inflammatory or stress-related proteins (interleukin (IL)-1ß, tumor necrosis factor (TNF)α, IL-2) [6]. The role of GDF-15 during atherosclerotic processes is controversially discussed. Already known is that, stimulation of MΦ with oxLDL leads to increased GDF-15 expression, foam cell formation, apoptosis and affects autophagic processes [8,9].

Autophagy is a highly evolutionarily conserved mechanism for the recycling and degradation of cytoplasmic constituents and is important for numerous physiological and pathological processes [10,11]. However, the role of autophagy in the development and progression of atherosclerosis seems to be complex, and, in the advanced stages of atherosclerosis, autophagy is impaired [12]. Moreover, clinical studies have shown that autophagy markers are localized in MΦ of atherosclerotic plaques [12,13].

Little is known about the regulation and mechanism of autophagy, especially in respect of GDF-15 and the pathogenesis of atherosclerosis [9]. Here, we proved the effect of GDF-15 of autophagy by using, in vitro, a human atherosclerosis MΦ model and in vivo GDF-15-deficient (GDF-15^−/−^) mice within an experimental atherosclerosis apolipoprotein (apo)E knockout (ApoE^−/−^) mouse model. In vitro, we show that GDF-15 regulates autophagic activity in human MΦ. In vivo, we show in the GDF-15-deficient setting that the autophagic processes in ECs of atherosclerotic plaque in the brachiocephalic trunk (BT) are regulated by GDF-15 after 20 weeks of feeding on a cholesterol-enriched diet (CED).

## 2. Materials and Methods

### 2.1. Cell Culture, Transfection and Gene Silencing

The human leukemic monocyte cell line THP-1 (Leibniz Institute DSMZ, Braunschweig, Germany) was used. Originally, the culture was derived from the blood of a one-year-old boy with acute monocytic leukemia, and, today, is frequently used as a model of monocyte/MΦ cell lineage [14]. THP-1 cells were cultured in RPMI-1640 medium (Capricorn Scientific GmbH, Ebsdorfergrund, Germany) and supplemented with penicillin and streptomycin (Capricorn Scientific GmbH) and 10% fetal bovine serum (Capricorn Scientific GmbH). Cells were cultured at 37 °C in a 5% CO_2_ environment, with a medium change every 2–3 days. All experiments were performed using cells at passage 9 and lower. THP-1 cells were differentiated into MΦ using 160 nM Phorbol 12-mystriate 13-acetate [PMA, (Sigma-Aldrich Chemie GmbH, Munich, Germany)] in RPMI 1640 medium for 72 h. Transfection of THP-1 MΦ with 50 nM siRNA for GDF-15 (FlexiTube GeneSolution GS9518, QIAGEN GmbH, Hilden, Germany) and with negative siRNA (siControl) (AllStars Negative Control, QIAGEN GmbH) was performed using HiPerfect Transfection Reagent (QIAGEN GmbH) following the manufacturer’s instruction. The efficacy of GDF-15 silencing has been recently published [9] (Appendix A). Because of the biological activity of rGDF-15 ED_50_ = 1.0–3.0 μg/mL (ProVitro, Berlin, Germany), PMA-differentiated THP-1 MΦ (3.5 × 10^6^ cells/mL) were treated with 1.0–1.5 μg/mL human rGDF-15 [9] (Appendix A).

### 2.2. Autophagic Activity Assay

To measure the autophagic activity in siGDF-15 or siControl THP-1 MΦ or after treatment with recombinant (r)GDF-15, cells were seeded (3 × 10^5^ cells/mL) in Lumox^®^ 96-well plates (Sarstedt AG & Co. KG, Nümbrecht, Germany). The Cell Meter™ Autophagy Assay Kit (AAT Bioquest^®^ Inc., Sunnyvale, CA, USA) includes the Autophagy Blue™ reagent, a specific autophagosome marker, to analyze the autophagic activity, according to manufacturer’s instructions. Thereafter, the cells were incubated (37 °C; 5% CO_2_; 20 min) and washed with buffer. The fluorescence intensity was determined at wavelength Ex/Em_333/518nm_, using the Cytation™ 3 microplate reader (BioTek Instruments GmbH, Bad Friedrichshall, Germany).

### 2.3. Lysosomal Activity Assay

The lysosomal activity was measured in PMA-differentiated THP-1 MΦ after GDF-15-silencing or treatment with rGDF-15, cells were seeded (3 × 10^5^ cells/mL) in Lumox^®^ 96-well plates (Sarstedt AG & Co. KG). The dye LysoBrite™ Red (AAT Bioquest^®^, Inc.) was used to measure lysosomal intracellular activity. The dye working solution was prepared by diluting 20 µL of the 500× LysoBrite™ stock in 10 mL RPMI medium without Phenol Red and without l-Glutamine (RPMI-Ly). After exposure of THP-1 MΦ to rGDF-15 for 4 h, the supernatant was removed and 100 µL/well of the working solution was applied. Thereafter, the cells were incubated (37 °C, 5% CO_2_; 30 min) and washed twice with RPMI-Ly. The fluorescence intensity was determined at wavelength Ex/Em_575/605 nm_ using the Cytation™ 3 microplate reader (BioTek Instruments GmbH).

### 2.4. Transmission Electron Microscopy (TEM)

For TEM, MΦ were fixed in Ito–Karnovsky mixture (2.5% glutaraldehyde, 2.5% paraformaldehyde, 0.05% picric acid) and washed in 100 mM cacodylate buffer (pH 7.3), afterwards were post-fixed (1% OsO_4_) and dehydrated in ethanol/propylene oxide. Thereafter, the samples were embedded in glycid ether 100 (EPON 812, SERVA Electrophoresis GmbH, Heidelberg, Germany). Ultrathin sections (65–80 nm) were cut using a Reichert Ultracut S ultramicrotome (Leica Microsystems, Wetzlar, Germany), mounted on copper grids and contrasted with 4% uranyl acetate and lead citrate. The specimens were observed with a Zeiss EM 10 C TEM (Carl Zeiss GmbH, Jena, Germany), and pictures were taken with Image a SP System (SYSPROG, Minsk, Belarus) [15,16].

### 2.5. SDS-PAGE and Western Blot

PMA-differentiated THP-1 MΦ were washed in ice-cold phosphate-buffered saline (PBS) and lysed using a radioimmunoprecipitation assay (RIPA) buffer of pH 7.5 (Cell Signaling Technology, Frankfurt, Germany), containing a protease/phosphatase inhibitor cocktail (Cell Signaling Technology). The protein concentrations were determined spectrophotometrically using the Pierce BCA (bicinchoninic acid) Protein Assay (Thermo Scientific, Rockford, IL, USA). Proteins were loaded on Novex™ WedgeWell™ 14%, Tris-Glycine (Thermo Scientific). Proteins were transferred onto 0.45 µm nitrocellulose membranes (Millipore, Billerica, MA, USA). Primary antibodies (Appendix A) were added and incubated overnight at 4 °C in blocking buffer (5% fat-free milk). Membranes were incubated with enhanced ECL-anti-rabbit IgG-POD antibody. The peroxidase reaction was visualized by AceGlow chemiluminescence substrate (PEQLAB GmbH, Erlangen, Germany) and was documented by the Fusion-SL Advance™ imaging system (PEQLAB GmbH) according to its instruction manual. The intensities of the specific Western blot bands were quantified using the software ImageJ, from the National Institutes of Health (Bethesda, MD, USA) and normalized against α-tubulin.

### 2.6. GDF-15 ELISA

The intracellular level of human GDF-15 in THP-1 MΦ was quantified by the DuoSet^®^ ELISA Development System (R & D Systems, Inc., Abingdon, UK). The capture antibody was coated to a 96-well MaxiSorp-ELISA Microplate (Nunc, San Diego, CA, USA) and incubated overnight at room temperature. According to manufacturers’ instructions, after the blocking step, the samples (2.5 μg protein/well) or standards were added to the well. After incubation with a detection antibody and streptavidin-HRP, we added the substrate solution (SigmaFast™ OPD, Sigma-Aldrich Chemie GmbH) to each well and there incubated for 30 min in the dark. The reaction was stopped with 50 μL 3 M HCl. The GDF-15-protein level (pg/mL) was measured with an ELISA reader (Tecan Deutschland GmbH, Crailsheim, Germany) at OD_490/655nm_.

### 2.7. Animals

GDF-15 knockout/lacZ knockin (GDF-15^−/−^) mice [17] were crossbred with ApoE^−/−^ mice (Charles River, Sulzfeld, Germany) to generate GDF-15^−/−^/ApoE^−/−^ mice. Male homozygous GDF-15^−/−^/ApoE^−/−^ and ApoE^−/−^ mice were used for this study and described by Bonaterra et al. [18]. At the age of 10 weeks, GDF-15^−/−^/ApoE^−/−^ and ApoE^−/−^ mice were fed for 20 weeks with an adjusted-calories cholesterol-enriched diet [CED; “western-type diet” (21% fat, 0.15% cholesterol and 19.5% casein), Altromin GmbH, Lage, Germany]. All animals had ad libitum access to food and water and appropriate environmental enrichment. The procedures were approved by the Regional Commission Gießen (V54-19c2015h01MR20/26Nr.G40/2017; 9.10.2017) and were performed in compliance with the regulations for animal experiments at the Philipps-University Marburg.

### 2.8. Genotyping

Genomic DNA was isolated from mouse ears using a commercial kit (DNA Extraction Solution; PeqLab, VWR Company, Erlangen, Germany) according to the manufacturer’s instructions (DirectPCR^®^ lysis reagent ear; Peqlab, VWR International). Subsequently, homozygous transgenic mice were detected by polymerase chain reaction (PCR) (Appendix A) using intron-spanning oligonucleotides (Eurofins Genomics, Ebersberg, Germany) as previously published [18].

### 2.9. Dissection and Tissue Harvesting

At the age of 30 weeks, the mice were weighed, narcotized and analgized with a combination of ketamine (150 mg/kg) and xylazine (20 mg/kg). Local intercostal anesthesia was performed with lidocaine 2%. The thoracic cavity was opened, the apex of the left ventricle was incised and a cannula (8 G, B. Braun Melsungen AG, Melsungen, Germany) was introduced and clamped. After the right atrial incision, the vascular system was perfused with a solution consisting of PBS with 5 UL/mL heparin (Liquemin^®^ 25,000 UL/5 mL, Roche, Grenzach, Germany), with a delivery volume of 30 mL and a rate of 100 mL/h, using an automated syringe-pump (Secura, B. Braun, Melsungen AG). Afterwards, 250–300 μL 0.9% NaCl sterile physiological solution containing methylene blue (0.25%; Riedel-de Haën, Seelze-Hannover, Germany) was injected into the vascular system. The blue-colored BT was excised under direct observation through a binocular loupe embedded in Tissue-Tek^®^ (Sakura Finetek, Stauffen, Germany) and snap-frozen in liquid nitrogen-cooled isopentane. Immediately, after opening the right atrium, heparinized (0.25 I.U./mL, Roche) blood samples were taken. Plasma was obtained by centrifugation (10 min, 650× *g*), stored at −80 °C. Plasma cholesterol, triglyceride (TG) as well as high-density lipoprotein (HDL) and low-density lipoprotein/very low-density lipoprotein (LDL/VLDL) levels were analyzed spectrophotometrically by using commercially available kits (Total Cholesterol/Cholesteryl ester quantitation assay kit, TG quantification assay kit and Cholesterol kit—HDL and LDL/VLDL, Abcam, Cambridge, UK) in a Cytation™ 3 microplate reader (BioTek Instruments GmbH). Body size was determined by measuring nasal-to-anal length, and body mass index (BMI) was calculated as the ratio between body weight and surface area (g/cm^2^) [19]. The body surface area was derived from the DuBois equation: body surface (m^2^) = 0.007184 × weight (kg^0.425^) × body size (cm^0.725^) [19].

### 2.10. Morphometry and Immunohistology

For morphometrical and immunohistological investigations, cryo cross-sectional series (6 μm) of the BTs were performed. The extent of atherosclerotic plaques in the BT was measured by computer-assisted morphometry. These images were evaluated and quantified with the software Fiji [20]. Standard hematoxylin-eosin (HE) and van-Giesson-Elastica staining were performed. Immunohistochemical staining method was carried out using the antibodies listed in Table 1. Nuclear counterstaining was performed by using 1 µg/mL DAPI (Sigma-Aldrich). The extent of the atherosclerotic lesions was determined by tracing of lumen and plaque areas along the internal elastic lamina (respectively luminal plaque circumference) and calculating [(plaque area [μm^2^])/(lumen area [μm^2^]) × 100% = lumen stenosis (%)]. The media was determined by tracing the area of the lumen along the internal elastic lamina and the area along the external elastic lamina by calculating: (luminal area to external elastic lamina [μm^2^]) − (luminal area to internal elastic lamina [μm^2^]) = area of media (µm^2^)]. Quantification of immunoreactive plaque areas was assessed [(immunoreactive plaque area [μm^2^])/(total plaque area [μm^2^]) × 100% = immunoreactive plaque area (%)] on the basis of Fiji [21]. The p62-accumulation of atherosclerotic plaque was analyzed by counting all p62-accumulation per plaque or cell type in the plaque area [(p62-accumulation)/DAPI) × 100% = p62 accumulation (%)] [9].

### 2.11. Statistical Analyses

Statistical analyses were performed using SigmaPlot 12 (Systat Software Inc., San José, CA, USA). After testing for normality (by Shapiro-Wilk), the unpaired Student’s t-test or one-way analysis of variance (ANOVA) was used. Data are reported as mean + standard error of the mean (SEM), and *p* ≤ 0.05 was considered as statistically significant.

## 3. Results

### 3.1. GDF-15 Silencing Reduces Autophagic Activity in Human THP-1 MΦ

Based on previously published data of the transfection efficacy with siGDF-15 RNA in PMA-differentiated human THP-1 MΦ [9], we performed autophagic activity assays in combination with lysosomal activity assays after siRNA transfection. After transient siGDF-15 transfection, we found a 22% (*p* = 0.04) decrease in autophagic activity in comparison with siControl MΦ (Figure 1a). Lysosomal activity showed no significant changes in siGDF-15 MΦ compared with siControl MΦ (Figure 1b). Serum starvation was used as positive control for an induction of autophagic and lysosomal activities as described by others [13]. After starvation for 4 h human THP-1 MΦ showed a 34% (*p* = 0.03) increase in autophagic activity (Figure 1a) and 18% (*p* = 0.01) increase in the lysosomal activity (Figure 1b) compared with siControl MΦ. Using TEM after starvation, THP-1 MΦ showed numerous vacuoles in the cytoplasm, depletion of organelles, as well as the presence of large autophagosomes containing membranous whorls and remnants of cytoplasmic material. These findings were less observed in siControl MΦ and siGDF-15 THP-1 MΦ (Figure 1c–e).

### 3.2. rGDF-15 Enhances Autophagic Activity in Human THP-1 MΦ

To investigate the effect of exogenous GDF-15 on MΦ autophagy, additional experiments were performed using rGDF-15. After incubation of PMA-differentiated human THP-1 MΦ with 1.0 μg/mL or 1.5 μg/mL rGDF-15 for 4 h, an autophagic (Figure 2a) and lysosomal activity assays (Figure 2b) were performed. Human THP-1 MΦ exposed to 1.0 μg/mL or 1.5 μg/mL rGDF-15 showed significant enhanced autophagic activity by about 20% (*p* = 0.04) and 16% (*p* = 0.02) compared with negative controls (~0 µg/mL GDF-15) (Figure 2a). The lysosomal activity in human THP-1 MΦ was not influenced upon exposure to rGDF-15 (Figure 2b). Using TEM, human THP-1 MΦ treated with rGDF-15 showed numerous vacuoles in the cytoplasm and the occurrence of large autophagosomes containing membranous whorls and remnants of cytoplasmic material (Figure 2c–e).

Additionally, Western blot analysis revealed that 1.5 µg/mL rGDF-15 treatment in human THP-1 MΦ resulted to an increased LC3-II/LC3-I ratio by enhanced LC3-II protein level (Figure 2f). The quantitative data revealed that 1.5 µg/mL rGDF-15 significantly increased the LC3-II/LC3-I ratio, by 104% (*p* = 0.038), compared with the negative control (taken as 100%, ~0 µg/mL GDF-15) (Figure 2g). The LC3-II expression was increased 43% (*p* = 0.045) compared with the negative control (Figure 2f,h). Treatment with 1.0 µg/mL rGDF-15 resulted to an enhanced LC3-I protein level around 47% (*p* = 0.034) compared with the negative control (Figure 2f,i). A starvation period of 4 h by human THP-1 MΦ, used as positive control, showed an increased LC3-II/LC3-I ratio (32%; *p* = 0.029) (Figure 2g) among increased LC3-I (77%; *p* = 0.049) and LC3-II protein levels (88%; *p* = 0.05) (Figure 2f–i).

### 3.3. GDF-15^−/−^ Reduces the Progression of Atherosclerotic Lesions in the BT of ApoE^−/−^ Mice

To address the question whether GDF-15 deficiency affects the development of atherosclerotic plaques in vivo, we investigated cross-sections of the BT in GDF-15^−/−^/ApoE^−/−^ and ApoE^−/−^ mice by histomorphometrical analyses (Figure 3). After 20 weeks feeding CED, GDF-15^−/−^/ApoE^−/−^ mice showed a significant decrease (*p* = 0.05)—about 25%—in maximal lumen stenosis in the BT compared with ApoE^−/−^ mice (Figure 3b–d). The area of the tunica media was not affected by GDF-15 deficiency in BT (Figure 3e).

### 3.4. GDF-15^−/−^ Mice Characterized by Increased Body Weight, BMI and Increased Plasma TG Level

Because of the reduced lumen stenosis in BT of GDF-15^−/−^/ApoE^−/−^ mice after CED, we investigated whether GDF-15 deficiency affects obesity and blood lipid concentration. GDF-15-deficient mice showed a significantly (*p* < 0.001, 44%) increased body weight and 23%-increased BMI after 20 weeks of CED (Table 2). To evaluate whether the gain of weight and BMI in GDF-15^−/−^/ApoE^−/−^ mice was due to changes in lipid metabolism, plasma TG, TC, FC, CE, HDL and LDL/VLDL levels were determined (Table 2). Twenty weeks of CED feeding resulted in an increased TG level of about 53% (*p* = 0.07) in GDF-15^−/−^/ApoE^−/−^ compared with ApoE^−/−^ mice (Table 2). Interestingly the lipid plasma levels of TC, CE, FC, HDL and LDL/VLDL were equal in GDF-15^−/−^/ApoE^−/−^ mice compared with ApoE^−/−^ after 20 weeks CED (Table 2).

### 3.5. GDF-15 Deficiency Decreased the Accumulation of p62 in Atherosclerotic Lesions, Especially in ECs

Next, we investigated the cellular structure in atherosclerotic lesions (Figure 4). The immunoreactive area of SMCs (sm-α-actin+), ECs (CD31+) and MΦs (CD68+) in atherosclerotic lesions were unaffected in GDF-15^−/−^ mice (Figure 4a–c). Additionally, the immunoreactive areas of autophagy (ATG5)-relevant proteins in atherosclerotic plaques were similar in GDF-15^−/−^/ApoE^−/−^ and ApoE^−/−^ mice (Figure 4d).

We investigated the accumulation of the autophagic marker p62 in atherosclerotic lesions (Figure 5). The accumulation of p62 was 57% (*p* = 0.002) decreased in atherosclerotic plaques of GDF-15^−/−^/ApoE^−/−^ compared with ApoE^−/−^ mice after 20 weeks CED (Figure 5a). Using double-immunofluorescence microscopy, we assigned p62 accumulation to specific cells in the atherosclerotic plaques (Figure 5b–d). Interestingly, the p62 accumulation in ECs (CD31+) was 61% (*p* ≤ 0.05) decreased in atherosclerotic plaques of GDF-15^−/−^/ApoE^−/−^ compared with ApoE^−/−^ mice after 20 weeks CED (Figure 5c), whereas the p62 accumulation in sm-α-actin and CD68+ MΦ were similar in GDF-15^−/−^/ApoE^−/−^ mice and ApoE^−/−^ mice (Figure 5b,d).

### 3.6. GDF-15 Deficiency Increased the Survivin Expression in Atherosclerotic Lesions, Especially in ECs

Impairment of autophagy in ECs promotes both apoptosis and senescence [22]. Therefore, we investigated survivin, which prevents apoptotic cell death, and p53 in CD31+ cells in atherosclerotic plaque of the BT of ApoE^−/−^ as well as GDF-15^−/−^/ApoE^−/−^ mice. The survivin+ plaque area was 2.2-fold (*p* = 0.003) increased in atherosclerotic plaques of GDF-15^−/−^/ApoE^−/−^ compared with ApoE^−/−^ mice after 20 weeks CED (Figure 6a,c,d). Using double-immunofluorescence microscopy, we found survivin accumulation in CD31^+^ cells in the atherosclerotic plaques (Figure 6b,e,f). The survivin accumulations in CD31^+^ cells was 1.7-fold (*p* = 0.035) increased in atherosclerotic plaques of GDF-15^−/−^/ApoE^−/−^ compared with ApoE^−/−^ mice after 20 weeks CED (Figure 6b). The p53^+^ plaque area was similar in GDF-15^−/−^/ApoE^−/−^ and ApoE^−/−^ mice after 20 weeks CED (Figure 7a–c). Using double-immunofluorescence microscopy, the CD31^+^ cells showed no positive p53-staining (Figure 7d,e).

## 4. Discussion

GDF-15 has been hypothesized to play a crucial role in cardiovascular diseases [8,23,24], especially in the development of atherosclerosis [18,25,26,27]. Previous studies shown, an increased level of GDF-15 is associated with the development and progression of atherosclerotic plaques, possibly through the regulation of autophagic and apoptotic processes [8,9]. Our previous data show that GDF-15 plays an important role during “basal or cytoprotective” autophagy, an initial barrier against stress induced-apoptosis [9]. Increased stress conditions, e.g., oxidative stress, lead to the induction of apoptosis, which results in the damage of cytoprotective mechanisms by cleavage of essential autophagy-related (ATG) proteins [28] and in consequence, inhibition of the autophagic flux [29,30]. Moreover, oxidative stress also induces GDF-15 expression [8]. Therefore, our present study focused on the role of GDF-15 on autophagic processes because both are predominantly involved in the development and progression of atherosclerotic lesions. For detailed investigations, we used an in vitro human atherosclerosis MΦ model, and in vivo GDF-15^−/−^/ApoE^−/−^ mice under CED [18], to study the role of GDF-15 on autophagic processes in atherosclerotic lesions. Therefore, we used different analyses (activity assays, TEM) and autophagy-relevant markers (ATG5, p62) to investigate and verify the influence of GDF-15 on the autophagic pathway in atherosclerosis.

In any case, TEM is a useful method to identify the autophagic process, because the formation of autophagic vacuoles is by far the most important morphological feature of autophagic cells. Autophagy starts with the sequestration of portions of the cytoplasm by a phagophore (special double-membrane structure), which matures into the autophagosome. Our present data show an increasing degree of concentric membranous lamellar formations “myelin figures” and represent the autophagic degradation of membranous cellular components, in human THP-1 MΦ, after starvation or after rGDF-15 treatment. After rGDF-15 treatment and especially after starvation terms we detected numerous late/degradative autophagic vacuoles or autolysosomes, with a typical one limiting membrane and electron-dense cytoplasmic material and/or organelles at various stages of degradation [31,32]. However, the membranous lamellar formations are not frequently present in human plaques but can be found in lesions of cholesterol-fed rabbits [33] or after treatment of SMCs in culture with oxidized lipids [34]. Additionally, we showed in nsiGDF-15 and siGDF-15 MΦ fewer autophagosomes, with the typical double-membrane [29], then in rGDF-15 treated MΦ.

These morphological results were confirmed by the autophagic activity by using the auto-fluorescent substance Monodansylcadaverine (MDC), a useful marker for late-stage autophagosomes and autophagolysosomes [29,35,36,37,38,39] and the lysosomal activity by LysoBrite™ Red, another fluorescent substance that selectively accumulates in lysosomes and can be used as evidence of autophagic cargo delivery to lysosomes [29]. Our data show that the autophagic activity in siGDF-15 MΦ was decreased, whereas rGDF-15 had an opposite effect. Interestingly, lysosomal activity was not affected upon GDF-15 treatment. The proper usage and functionality of the autophagic and lysosomal activities were evaluated, using starvation as a positive control.

Another method to monitor autophagy machinery is western blot with the specific marker LC3. LC3 is initially synthesized in an unprocessed form (proLC3). Due its lack of amino acids on the C terminus, proLC3 is converted into LC3-I, a proteolytically processed form. Finally, LC3-I was modified into the PE-conjugated form, LC3-II. The LC3 conversion from cytosolic soluble form (LC3-I) to membrane-bound form (LC3-II) is the key issue during autophagosome formation [40,41]. Autophagic flux is the final stage, in which the autophagosome is digested by lysosomes such that the LC3-II protein will be partially decreased. During this stage, if the autophagic flux was disturbed, especially by lysosomal protease inhibitors such as E64d and pepstatin A, LC3-II increased. Mouse embryonic fibroblasts (MEFs) show, during starvation, a decreased protein level of LC3-I, in which LC3-II protein level increased [42]. In the present study we demonstrated that rGDF-15 induce LC3 conversion and resulted in an increased LC3-II protein level. The amount of LC3-II is closely correlated with the number of autophagosomes and corroborates the process of autophagosome formation [40]. Our current results are consistent with a previous study, which showed that GDF-15 silencing in MΦ (siGDF-15 MΦ) led to an impaired ATG5, ATG12/ATG5-complex and p62-protein level, as well as lesser p62 accumulations compared with nsiGDF-15 MΦ, whereas rGDF-15 promoted p62 accumulation [9]. Based on this previous data, the current study verifies the influence of GDF-15 on autophagic activity with consequences on p62 turnover in human THP-1 MΦ. One important step of autophagy is the autophagosome-lysosome fusion, which leads to lysosomal degradation of the sequestered materials by various lysosomal hydrolytic enzymes [43]. This process of “autophagic-flux” can be documented by investigating the autophagy receptor p62, which is a marker of autophagic status [43,44,45,46]. Defect autophagy increases the quantity of p62, a protein that, if overexpressed, can stimulate the production of reactive oxygen species and cell death [44]. In relation to any GDF-15 effects on the lysosomal activity, we imply that GDF-15 promotes apoptotic processes in human MΦ as a consequence of increasing autophagic activity when the autophagic cargo delivery to lysosomes (lysosomal activity) is constant, which will reach an impaired autophagy flux with increased p62-accumulation [9]. Therefore, in this study, we further proved the influence of GDF-15 on autophagy by using relevant markers, like p62, in the context of plaque progression by using a GDF-15^−/−^/ApoE^−/−^ mouse model.

Adult GDF-15^−/−^/ApoE^−/−^ mice showed increased body weight and BMI compared with the ApoE^−/−^ genotype. The effect of GDF-15 deficiency on body weight corresponds with previous data [18] and with the observation that transgenic mice, which overexpressed GDF-15, showed hypophagia as well as reduced body weight [26,47]. In accordance with previous observations, the triglyceridemia was elevated in mice lacking GDF-15, whereas the cholesterol level was unchanged after 20 weeks CED, compared with ApoE^−/−^ mice [18]. Additionally, the finding that GDF-15 deficiency considerably inhibits lumen stenosis, was previously described in the literature [18,27]. Therefore, we conclude that the inhibition of lesion progression in the absence of GDF-15 is obviously not due to a modification of plasma lipid levels. Consequently, other mechanisms or factors must be involved. One of these mechanisms could be autophagy. In this context, it was of high interest for this study to address the question of whether GDF-15 influenced cell death processes such as autophagy.

At first, we investigated the cellular composition and distribution in atherosclerotic plaque, because GDF-15 has been supposed to be involved in orchestrating atherosclerotic lesion progression [18]. Immunohistomorphometric analyses of atherosclerotic lesions in the BT show that GDF-15 deficiency did not affect the percentage of the sm-α-actin^+^ area, CD31^+^ area (EC), CD68^+^ area (MΦ), as well as ATG5^+^ area (autophagy) in atherosclerotic plaques after 20 weeks CED. Other studies showed, by immunohistological investigation, a significant reduction in the number of APG5L/ATG^+^ cells in GDF-15-deficient mice after 20 weeks of CED [18]. We suppose that the discrepancies may be due to the use of different staining methods.

The central findings of the previous observation suggest downregulation of apoptotic and autophagic processes in the absence of GDF-15 [18] because both apoptosis and autophagy are involved in cell death processes. Blocking autophagy renders MΦ in apoptosis, worsens the recognition and clearance of the dead cells by efferocytosis, and promotes plaque necrosis in a mouse model of advanced atherosclerosis [48]. For that reason, we investigated the accumulation of p62 in atherosclerotic lesions, because p62 is a critical checkpoint of the extrinsic apoptosis pathway, to control cell death and/or survival [49]. p62-protein regulates cell survival by the packing and delivery of polyubiquitinated, misfolded, aggregated proteins and dysfunctional organelles [50,51,52]. Increased levels of p62 in atherosclerotic plaques likely reflect dysfunctional autophagy because this reflects defects during the fusion stage with the lysosomes [53]. Here we show that the accumulation of p62 was significantly decreased in atherosclerotic plaques of GDF-15^−/−^/ApoE^−/−^ mice after 20 weeks CED, as well as after the silencing of GDF-15, which leads to a reduction of p62 accumulation in THP-1 MΦ [9]. Moreover, Bonaterra et al. observed significantly fewer apoptotic cells in the atherosclerotic plaques of GDF-15^−/−^/ApoE^−/−^ mice after 20 weeks of CED compared with ApoE^−/−^ mice [18]. However, the induction of apoptosis results in the damage of cytoprotective mechanisms by the cleavage of essential ATG proteins [28,54] and consequently an inhibition of the autophagic flux, followed by increased p62 accumulation. Interestingly, only in CD31^+^ cells p62 accumulation significantly decreased in atherosclerotic plaques of GDF-15^−/−^/ApoE^−/−^ mice after 20 weeks CED. Grootaert et al. hypothesized that defective endothelial autophagy plays a direct role in aging-related arterial dysfunction, because defective autophagy in ECs leads to apoptosis and senescence [53], whereas and vice versa, functional endothelial autophagy limits plaque formation under high shear stress conditions [22].

Ionizing radiation (IR) induces mitochondrial ROS production in ECs and therefore causes damage and cellular senescence [55]. GDF-15, released in senescent ECs, contributes to the pathogenesis of atherosclerosis via its pro-senescent activity, implicating endothelial loss of function [55,56]. Also, senescent ECs expressed an increased level of GDF-15, whereas the paracrine effect of GDF-15 was associated with EC proliferation, migration and nitric oxide by non-senescent ECs [57]. Another study shows that GDF-15 causes endothelial dysfunction by impairing vascular contraction and relaxation [58]. Our study shows that GDF-15^−/−^/ApoE^−/−^ mice have increased survivin expression in atherosclerotic plaques, especially increased percentage of survivin positive ECs compared with ApoE^−/−^ mice. Survivin, also known as Birc5, is a member of an inhibitor of the apoptosis protein family [59]. The general function of survivin is to inhibit cell apoptosis and promote proliferation [60,61]. It was previously suggested that survivin is not expressed in the normal adult vascular wall of mice and rabbits [62]. Several publications indicate that survivin is a negative regulator of autophagy that interacts with different proteins of the autophagic machinery, such as LC3, and interferes in the formation of autophagosomes, preventing LC3-I’s cleavage into LC3-II. The survivin inhibitor YM155 increases the conversion of LC3-II and promotes autophagy-mediated ROS production, DNA damage and cell death in breast cancer cells [63,64]. Also, survivin inhibits the conjugation and complexation between ATG12, ATG5, and ATG16L1 which are crucial for the elongation of autophagophores during canonical autophagy [65]. It seems that survivin can interfere with the elongation of autophagosomes in ECs and prevent excessive autophagy, apoptosis and/or senescence following endothelial dysfunction in GDF-15^−/−^/ApoE^−/−^ mice after 20 weeks CED. On the other hand, we analyzed p53 in ECs of GDF-15^−/−^/ApoE^−/−^ and ApoE^−/−^ mice after 20 weeks CED. p53 protein induces apoptosis by regulating the expression of several apoptotic genes. In particular, p53 binds specific elements of the survivin promoter and represses survivin expression [66,67]. In our study, the expression of p53 in atherosclerotic plaque was not detectable in ECs of both mice genotypes. These findings may imply a linkage between survivin and GDF-15 in relation to autophagy and apoptosis in arteriosclerotic plaques.

Our study suggests that GDF-15 is involved in establishing atherosclerotic lesions by the regulation of autophagic processes, which may have important pathophysiological consequences for atherosclerotic plaque progression and, thus, may be useful in developing novel strategies for therapeutic intervention.

## Figures and Tables

**Figure 1 cells-10-02346-f001:**
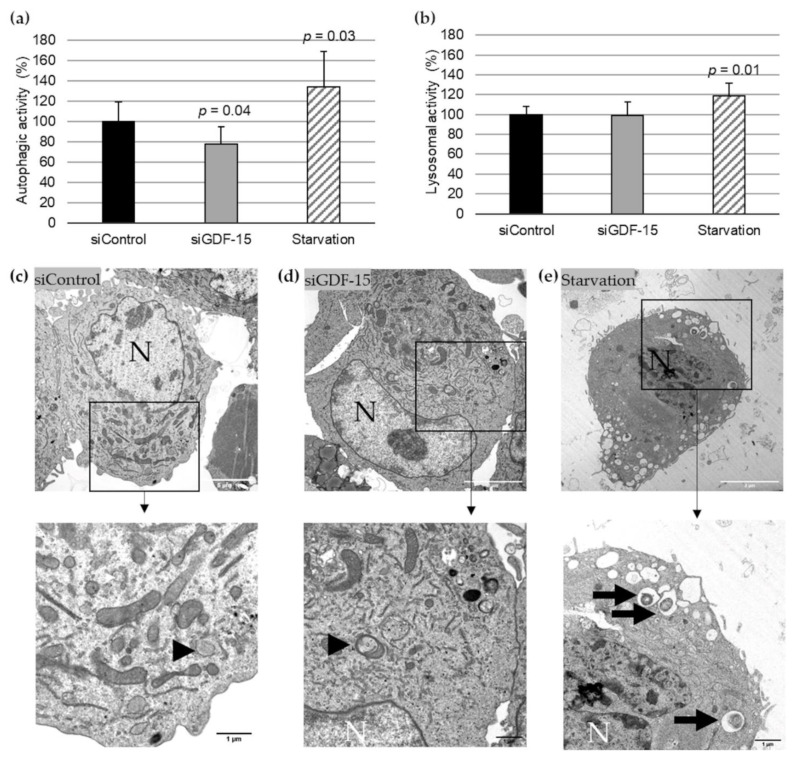
Autophagic and lysosomal activities in siControl and siGDF-15 THP-1 MΦ. (**a**) Autophagic activity (%) in human THP-1 MΦ (siControl, siGDF-15, starvation) was determined using fluorescent Cell Meter™ Autophagy Assay Kit [OD_333/518_]. (**b**) Lysosomal activity (%) in human THP-1 MΦ (siControl, siGDF-15, starvation) was determined using LysoBrite™ Red [OD_575/597_]. *n* = 4–5 independent experiments. Data are presented as mean + SEM. nsiGDF-15 MΦ = 100%; (**c**–**e**) TEM of THP-1 MΦ. N, nucleus; black arrows: autophagic vacuoles/autolysosomes; black arrowhead: autophagosome. White scale bars: 5 µm; black scale bars: 1 µm.

**Figure 2 cells-10-02346-f002:**
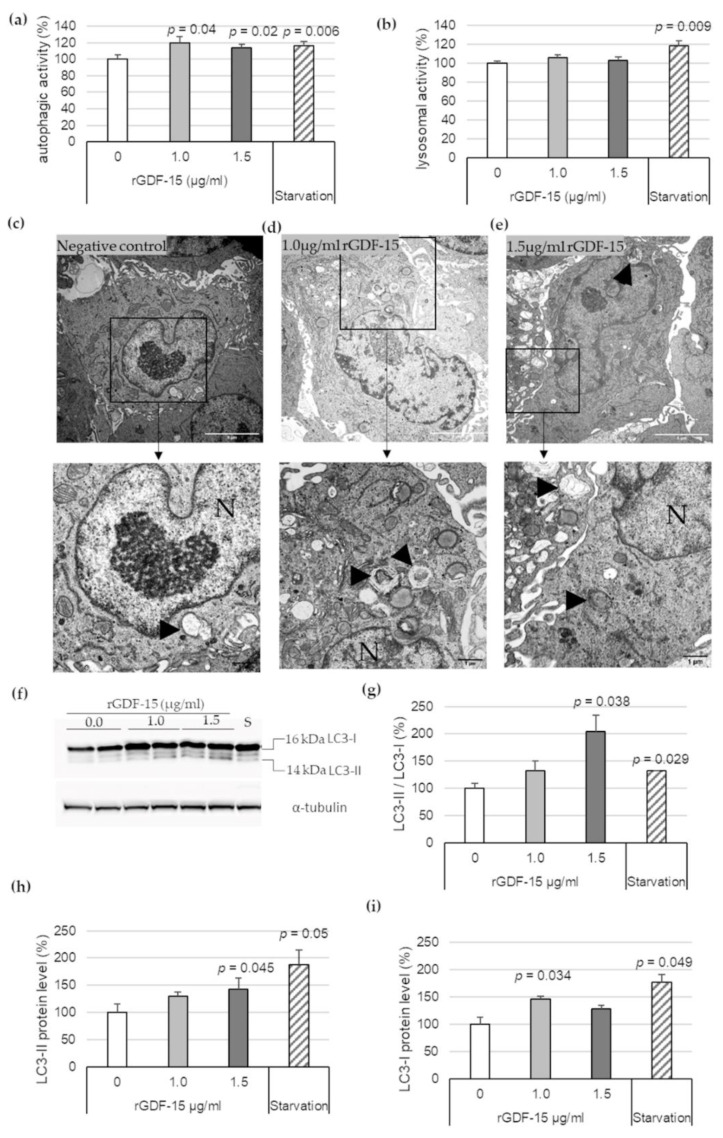
Autophagic and lysosomal activities in rGDF-15-treated human THP-1 MΦ. (**a**) Autophagic activity (%) in rGDF-15-treated human THP-1 MΦ were determined using fluorescent Cell Meter™ Autophagy Assay Kit [OD_333/518_]; (**b**) Lysosomal activity (%) in human THP-1 MΦ in rGDF-15-treated human THP-1 MΦ were determined using LysoBrite™ Red [OD5_75/597_]. Data are presented as mean + SEM. medium alone = 100%; (**c**–**e**) TEM of human THP-1 MΦ. N, nucleus; Black arrowhead autophagic vacuoles/autolysosomes. White scale bars: 5 µm; black scale bars: 1 µm. (**f**) Representative images of Western blot results for LC3-I, LC3-II and α-tubulin. S = starvation (**g**–**i**) percentage of (**g**) LC3-II/LC3-I ratio, (**h**) LC3-II and (**i**) LC3-I protein levels in THP-1 MΦ. Expression was normalized against α-tubulin and quantified by ImageJ. Data are presented as mean + SEM (four independent experiments were performed). 0 µg/mL rGDF-15 = 100%.

**Figure 3 cells-10-02346-f003:**
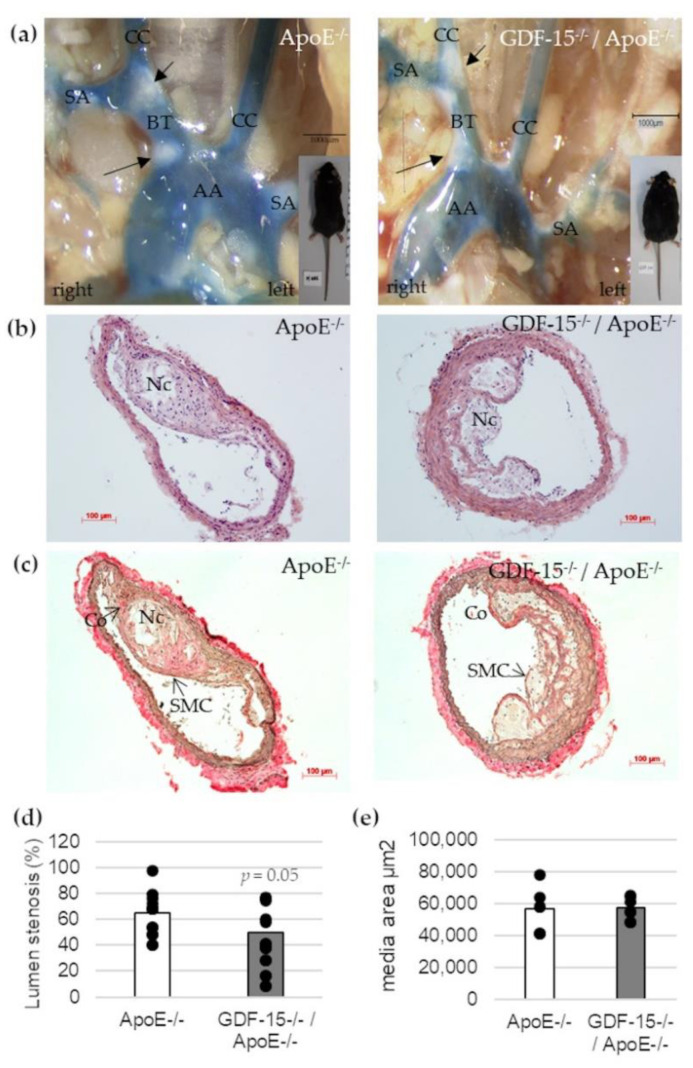
Effects of GDF-15 deficiency on lumen stenosis in the brachiocephalic trunk (BT) of ApoE^−/−^ and GDF-15^−/−^/ApoE^−/−^ mice after 20 weeks CED. (**a**) Open chest picture. AA, aortic arch; CC, common carotid artery; SA, subclavian artery; black arrows: atherosclerotic plaques. Scale bar: 1000 µm; (**b**) Representative HE stained and (**c**) Elastica-van Giesson-stained histological cross-sections of the BT of GDF-15^−/−^/ApoE^−/−^ and ApoE^−/−^ mice. Nc, necrotic core; Co, collagen fibers; SMC, smooth muscle cells; Scale bar: 100 µm; (**d**) Lumen stenosis (%) and (**e**) media area (µm^2^) were measured in BT by computer-assisted morphometry.

**Figure 4 cells-10-02346-f004:**
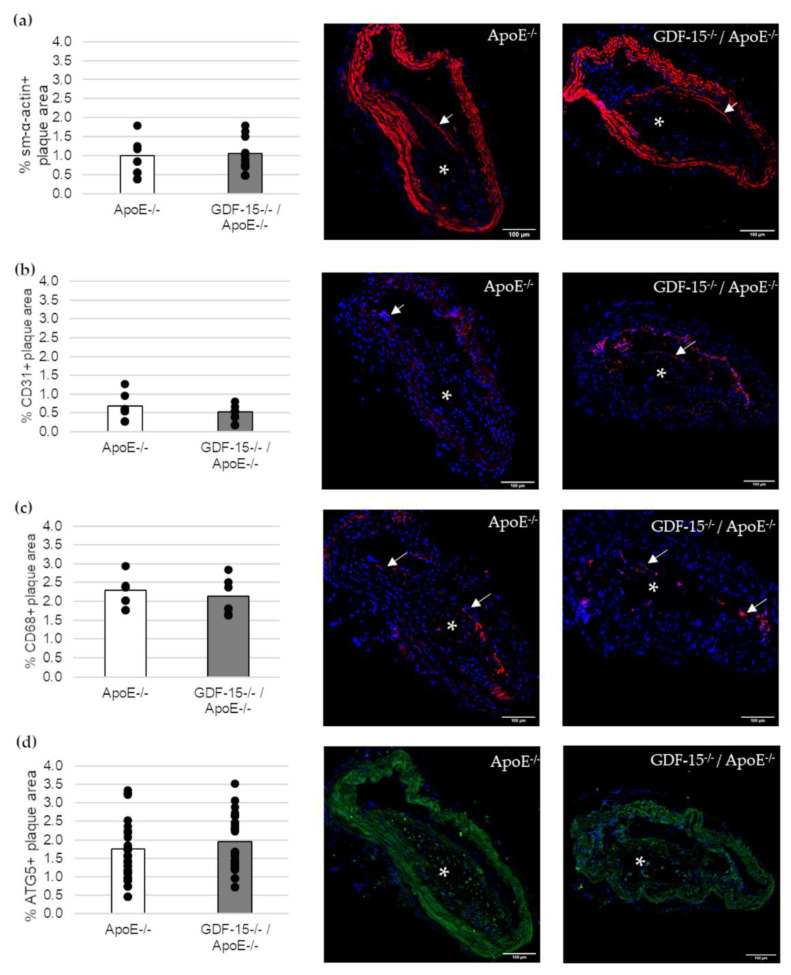
Immunofluorescence staining in an atherosclerotic plaque of the BT of ApoE^−/−^ and GDF-15^−/−^/ApoE^−/−^ mice after 20 weeks CED. (**a**) sm-α-actin immunoreactivity (ir) (*n* = 7–10); (**b**) CD31 ir (*n* = 6); (**c**) CD68 ir (*n* = 6–9); (**d**) ATG5 ir (*n* = 23). Nuclei are counterstained with DAPI. White arrows = positively stained cells. asterisk = atherosclerotic plaque; Scale bar: 100 µm. Positively stained plaque area (%) were measured by computer-assisted morphometry. Total plaque area = 100%. Scale bars: 100 µm.

**Figure 5 cells-10-02346-f005:**
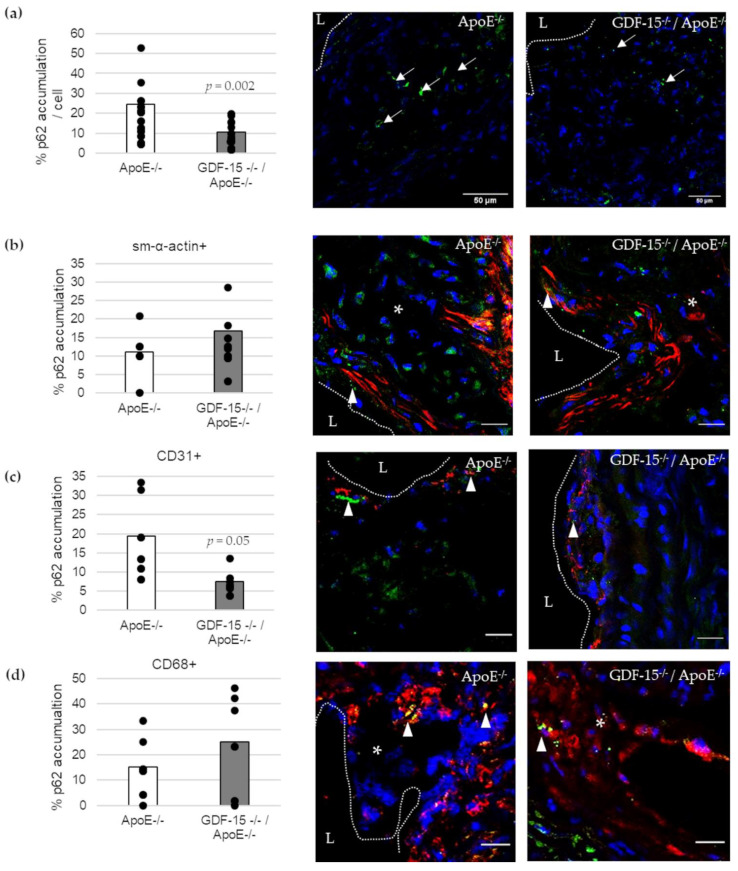
p62-accumulation and localization in an atherosclerotic plaque of the BT of ApoE^−/−^ and GDF-15^−/−^/ApoE^−/−^ mice after 20 weeks CED. (**a**) Percentage of p62-accumulation in an atherosclerotic plaque of the BT of GDF-15^−/−^/ApoE^−/−^ and ApoE^−/−^ mice. (white arrows, *n* = 15–21); (**b**–**d**) p62 accumulation are localized in (**b**) sm-α-actin ir, (**c**) CD31 ir and (**d**) CD68 ir cells as seen in the overlay (arrowheads, *n* = 5–10). Nuclei are counterstained with DAPI. p62 accumulation (%) and double-stained area (%) were measured by computer-assisted morphometry and normalized against DAPI.; L = arterial lumen; asterisk = atherosclerotic plaque; (**a**) scale bar = 50 µm; (**b**–**d**) scale bar = 20 µm.

**Figure 6 cells-10-02346-f006:**
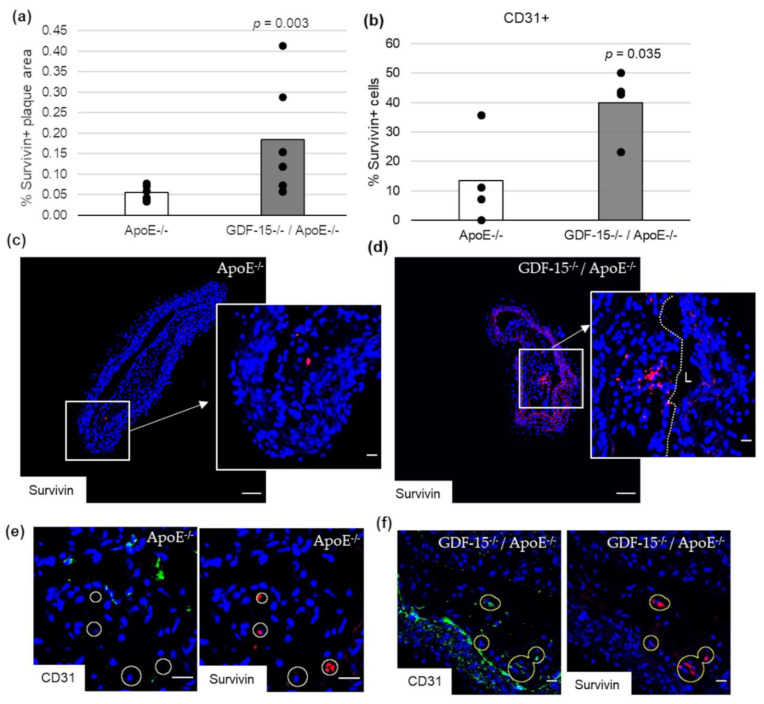
Survivin localization in atherosclerotic plaque of the BT of ApoE^−/−^ and GDF-15^−/−^ApoE^−/−^ mice after 20 weeks CED. (**a**) Percentage of survivin+ plaque area in atherosclerotic lesion of the BT of GDF-15^−/−^/ApoE^−/−^ and ApoE^−/−^ mice. (*n* = 4–6); Positively stained plaque area (%) were measured by computer-assisted morphometry. Total plaque area = 100%. (**b**) Percentage of survivin+ staining in CD31+ cells; normalized against DAPI (*n* = 4). (**c**,**d**) survivin ir (*n* = 4–6); (**e**,**f**) survivin are localized in CD31+ cells. Nuclei are counterstained with DAPI. Circle = district of survivin+ cells. L = arterial lumen; (**c**,**d**) overview screen: scale bar = 100 µm; (**c**–**f**) scale bar = 20 µm.

**Figure 7 cells-10-02346-f007:**
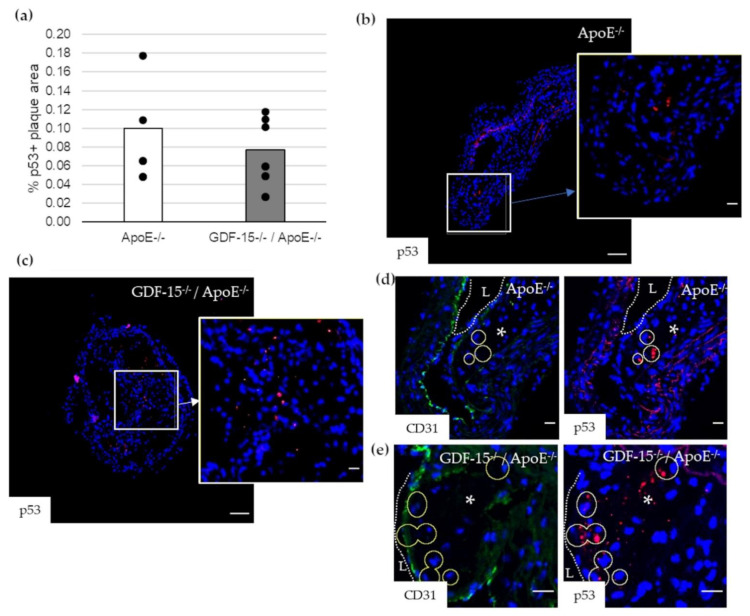
p53 localization in atherosclerotic plaque of the BT of ApoE^−/−^ and GDF-15^−/−^/ApoE^−/−^ mice after 20 weeks CED. (**a**) Percentage of p53+ plaque area in atherosclerotic plaque of the BT of GDF-15^−/−^/ApoE^−/−^ and ApoE^−/−^ mice. (*n* = 4–6); (**b**,**c**) p53 ir (*n* = 4–6); (**d**,**e**) p53 are localized in CD31^+^ cells. Nuclei are counterstained with DAPI. Circle = district of p53^+^ cells. L = arterial lumen; asterisk = atherosclerotic plaque. Positively stained plaque areas (%) were measured by computer-assisted morphometry. Total plaque area = 100%. (**b**,**c**) Overview screen: scale bar = 100 µm; (**b**–**e**) scale bar = 20 µm.

**Table 1 cells-10-02346-t001:** Antibodies used in this study.

Name	Cat Nr.	Company	Dilution
Primary Antibodies			
goat anti-mouse Smooth Muscle Actin (sm-α-actin)	ABIN185271	Antibodies online	1:10
rabbit anti-mouse APG5L/ATG5	ab108327	ABCAM, Cambridge, UK	1:10
rat anti-mouse CD68	MCA1957T	Bio-Rad Laboratories Inc., Hercules, CA, USA	1:10
rabbit anti-mouse p62/SQSTM1	P0068	Sigma-Aldrich Chemie GmbH, Munich, Germany	1:50
rat anti-mouse CD31	ab73888	ABCAM, Cambridge, UK	1:10
Survivin Alexa Fluor674	sc-17779 AF647	Santa Cruz Biotechnology, Heidelberg, Germany	1:50
p53 Alexa Fluor674	sc-126 AF647	Santa Cruz Biotechnology, Heidelberg, Germany	1:50
**Secondary Antibodies**			
goat Fab anti-rat IgG (H + L)-Cy3	112-167-003	Jackson ImmunoResearch, Ely, UK	1:100
rabbit IgG anti-goat IgG (H + L)-Cy3	305-165-003	Jackson ImmunoResearch, Ely, UK	1:100
goat anti-rabbit IgG (H + L)-Alexa Fluor 488	A-11008	Thermo Fisher Scientific, Rockford, IL, USA	1:200

**Table 2 cells-10-02346-t002:** Effects of GDF-15^−/−^ on body weight, BMI, plasma TG, cholesterol and HDL/LDL levels.

	ApoE^−/−^ (*n*)	GDF-15^−/−^/ApoE^−/−^ (*n*)
**Body Size (cm)**	9.77 ± 0.19 (6)	9.50 ± 0.32 (14)
**Body Weight (g)**	30.91 ± 2.21 (6)	44.44 ± 3.35 *** (14)
**BMI (g/cm^2^)**	3.61 ± 0.05 (6)	4.42 ± 0.06 *** (14)
**TG (mg/dL)**	135.66 ± 53.42 (6)	207.07 ± 78,81 (10)
**TC (mg/dL)**	440.37 ± 70.37 (6)	514.36 ± 92.11 (10)
**FC (mg/dL)**	126.42 ± 61.20 (6)	112.34 ± 36.32 (10)
**CE (mg/dL)**	313.94 ± 40.71 (6)	402.02 ± 106.99 (10)
**HDL (mg/dL)**	0.51 ± 0.58 (6)	1.58 ± 1.61 (10)
**LDL/VLDL (mg/dL)**	12.99 ± 4.23 (6)	13.67 ± 6.58 (10)

*** *p* < 0.001 vs. ApoE^−/−^ CED; mean ± SD. CED, cholesterol-enriched diet; GDF-15, growth-differentiation factor 15; BMI, body mass index; TG, triglyceride; TC, total cholesterol; FC, free cholesterol; CE, cholesterol ester; HDL, high-density lipoprotein; LDL/VLDL, low-density lipoprotein/very low-density lipoprotein.

## Data Availability

All relevant data are presented in this paper.

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
