# Peer review of "GDF-15 Deficiency Reduces Autophagic Activity in Human Macrophages In Vitro and Decreases p62-Accumulation in Atherosclerotic Lesions in Mice"

_cells, 2021, doi:10.3390/cells10092346_

Round 1

Reviewer 1 Report

The study by Heduschke investigated the role of GDF-15 in autophagic activity in atheroschlerotic lesions in mice and human macrophages in vitro. The authors used siRNA and recombinant proteins for GDF-15 loss and gain of function experiments and developed a novel mouse strain that combines the GDF-15 deficiency with well-established ApoE knockout phenotype. The authors reported experimental data supporting the important role of GDF-15 in the regulation of autophagy especially in endothelial cells. The study is compelling and very well elaborated and the conclusions are solid.

I have no major objections to the study.

Minor comments:

  1. Please address endothelial cell specific function of GDF-15 and its loss in these cells in more detail. The authors addressed in vitro the GDF-15 inhibition in macrophages. Why not in endothelial cells? Is a direct effect of GDF-15 siRNA on p62 levels in ECs to be expected, or is a reduction of GDF-15 levels in macrophages responsible for reduced levels of p62 in ECs? What would be the mechanism behind?

  1. What is the rationale of dosage choice used for stimulants (figure 2)? Have the authors titrated the concentration and why are these specific concentrations determined?

  1. The depiction of error bars is not necessary if “individual data plots" are shown (e.g. Figure 3d)

  1. Figure 5 c: please display the exact p value to stay consistent with other figures. (p=xxx)

  1. Figure 5: Display format of scale bars differ between images. Please provide consistent scale bars (you could easily remove the text from the scale bar).

Reviewer 2 Report

The authors demonstrated that GDF-15 gene silencing reduced autophagy in macrophages in vitro while administration of recombinant GDF-15 increased it. GDF-15-/- ApoE -/- fed with cholesterol-enriched diet (CED) showed reduced lumen stenosis and decreased p62 accumulation in endothelial cells.

The study is interesting and suggests that GDF-15 represents an important player in atherosclerosis progression, through the modulation of autophagy.

However, some comments should be addressed:

1) In addition to TEM and the other assays performed in this study, I encourage the authors to evaluate autophagy and autophagic flux in vitro with different techniques, as also recommended by the “Guidelines for the use and interpretation of assays for monitoring autophagy (4th edition)”. For example, western blot for LC3 and p62, in presence or in absence of lysosomal inhibitors should be performed in silenced cells and in cells treated with rGDF-14. mRFP-GFP-LC3 represents another valid option to monitor autophagy.

2) Autophagy evaluation in vivo also requires further analysis, for example the assessment of Beclin1 and LC3.

3) The role of endogenous GDF-14 is not clear. The authors should evaluate the expression of GDF-14 in macrophages in vitro or in Apo E -/- fed with CED.

4) Since the authors hypothesized a mechanistic link between autophagy deficiency, apoptosis and atherosclerosis progression, markers of apoptosis should be investigated.

Minor:

Title: REDUCES, lowercase letters

Reviewer 3 Report

This manuscript studies the role of GDF-15 in macrophage and argued that it’s in charge of autophagy regulation. It is important to understand the protective role of GDF-15 against atherosclerosis.

Unfortunately, the study design and the approaches are not convincing. 

One of major concerns is that the authors never showed the protein expressions of GDF15 and LC3. The study is about GDF-15 and claimed that the expression is manipulated. And LC3 or its family proteins (eg. GABARAP) are the most common and reliable autophagy marker. This study missed the evidence of validity of the study model.

Although autophagy assay kits are generally good tool for high-throughput screening, the accuracy is a little questionable. In addition, LysoBrite assay is just staining acidic compartments but does not guarantee the activity of lysosomal enzymes. Thus, the individual finding should be scrutinized by applying a comprehensive autophagy flux assay. Western blotting by using anti-LC3 antibody is one of standard assay method.

The evidence of efficacy and the specificity of siRNA targeting GDF15 should be presented by western blotting. The expression level of rGDF-15 should also be determined.

The p62 results are confusing. When p62 is used as an autophagy marker, the accumulation indicates the reduction of autophagy activity and the less expression suggests upregulation of autophagy. The results and the descriptions are against this general interpretation. p62 and ATG5 can support, but are not sufficient to measure autophagy activity. Please use LC3 as a maker.

More importantly, the presented data can not exclude the possibility that the accumulation of p62 and the increased signal in the autophagy assay are due to the failure of autophagosome-lysosome fusion. It is essential to demonstrate an appropriate autophagy flux measurement by using bafilomycinA1 or chloroquine. Please refer to a paper regarding this issue such (eg. Loos, Autophagy, 2014, PMID: 25484088).

There is no quantitative autophagy analysis. TEM and the other microscopic approaches are not suitable for quantitative analysis. This manuscript relies too much on the observational interpretation.

The method description of rGDF-15 preparation and the administration are unclear. It needs to describe detailed more.

The labeling "nsiGDF-15" is confusing, can be simply siControl.

Round 2

Reviewer 2 Report

No further comments

Reviewer 3 Report

Although the revised manuscript provides additional data, it is not sufficient to address the major concerns. The quality of LC3 western blotting is not satisfactory. The form II bands look too weak to obtain accurate measurements. In addition, as referred in the authors’ cover letter  (#40, Mizushima, 2007), to obtain valid autophagy flux, the experiments should be repeated under the condition of the presence and absence of lysosomal protease inhibitors. 

PCR was used to confirm the knockout model and ELISA assay was used for measuring GDF-15 expression. However, the efficacy of siRNA and rGDF15 are still unclear. Moreover, western blot against GDF-15 would be more convincing since that would allow to discuss the specificity of the antibody.